# Beta Distribution-Based Cross-Entropy for Feature Selection

**DOI:** 10.3390/e21080769

**Published:** 2019-08-07

**Authors:** Weixing Dai, Dianjing Guo

**Affiliations:** School of Life Science and State Key Laboratory of Agrobiotechnology, G94, Science Center South Block, The Chinese University of Hong Kong, Shatin 999077, Hong Kong, China

**Keywords:** cross-entropy, beta distribution, feature selection, machine learning, data mining

## Abstract

Analysis of high-dimensional data is a challenge in machine learning and data mining. Feature selection plays an important role in dealing with high-dimensional data for improvement of predictive accuracy, as well as better interpretation of the data. Frequently used evaluation functions for feature selection include resampling methods such as cross-validation, which show an advantage in predictive accuracy. However, these conventional methods are not only computationally expensive, but also tend to be over-optimistic. We propose a novel cross-entropy which is based on beta distribution for feature selection. In beta distribution-based cross-entropy (BetaDCE) for feature selection, the probability density is estimated by beta distribution and the cross-entropy is computed by the expected value of beta distribution, so that the generalization ability can be estimated more precisely than conventional methods where the probability density is learnt from data. Analysis of the generalization ability of BetaDCE revealed that it was a trade-off between bias and variance. The robustness of BetaDCE was demonstrated by experiments on three types of data. In the exclusive or-like (XOR-like) dataset, the false discovery rate of BetaDCE was significantly smaller than that of other methods. For the leukemia dataset, the area under the curve (AUC) of BetaDCE on the test set was 0.93 with only four selected features, which indicated that BetaDCE not only detected the irrelevant and redundant features precisely, but also more accurately predicted the class labels with a smaller number of features than the original method, whose AUC was 0.83 with 50 features. In the metabonomic dataset, the overall AUC of prediction with features selected by BetaDCE was significantly larger than that by the original reported method. Therefore, BetaDCE can be used as a general and efficient framework for feature selection.

## 1. Introduction

When a machine learning algorithm learns from high-dimensional data, a huge amount of training samples is typically required to ensure the predictive power. This problem is known as the “curse of dimensionality” [1]. Feature selection is, therefore, commonly used to control the dimensionality in order to make the machine learning model more robust and the data more interpretable. Feature selection is widely used in various fields such as economics [2], fault diagnosis [3], DNA microarrays [4], etc.

Feature selection algorithms can be broadly divided into three categories: filter, wrapper, and embedded methods [5]. Filter methods select features without any learning algorithms and evaluate features individually with evaluation criterion such as mutual information [6], pair *t*-scores [7], and relief-based methods [8]. By contrast, wrapper methods evaluate subsets of features by learning algorithms such as random forest (RF), support vector machine (SVM), k-nearest neighbor (KNN), and neural network (NN), together with searching algorithms such as sequential forward selection [9], genetic algorithm [10], and fuzzy C-means [11,12], to search for the optimal subset of features. Embedded methods are quite similar to wrapper methods except that an intrinsic building metric is used in the model during training. For example, the least absolute shrinkage and selection operator (LASSO) method uses a linear model to shrink many features to zero by L1 penalty. Other embedded methods include Bolasso [13], FeaLect [14], and support vector machine–recursive feature elimination (SVM-RFE) [15,16,17]. Among the three categories, filter methods are the most computationally efficient but usually have the worst performance. They are generally used as a preprocessing method for feature selection. Wrapper and embedded methods perform better than filter methods, but with a higher computational cost.

The evaluation function is the basis of feature selection that determines the quality of the finial models [18]. Different from filter methods, most methods of high performance, including wrapper and embedded methods, are based on evaluation of the accuracy by cross-validation [19,20]. Other evaluation functions based on information theory include Akaike information criterion (AIC) and Bayesian information criterion (BIC), which are derived via the maximum entropy principle [21,22] and minimum description length [23] respectively. AIC and BIC both have a term of accuracy, and the difference lies in the penalty for the number of features. Under some circumstances, AIC is asymptotically equivalent to leave-one-out cross-validation [24,25]. However, cross-validation itself can result in over-optimistic estimates and poor generalization ability, thus leading to poor performance for feature selection [26,27].

Many recent studies on feature selection focused on the methods of optimizing feature subsets. New searching methods were applied to select the optimal feature subset. For example, whale optimization-related approaches were used for wrapper feature selection [28]. Other new approaches include evolutionary population dynamics and grasshopper optimization [29], binary dragonfly optimization [30], artificial bee colony optimization [31], etc. Combined methods of optimization for feature selection were applied in several studies. For example, Mafarja et al. combined the whale optimization with simulated annealing [32]. Faris et al. combined binary salp swarm algorithm with crossover scheme for feature selection [33]. However, few studies were conducted on evaluation function, which is another important element for feature selection other than the searching method.

Hence, we propose a new algorithm, named beta distribution-based cross-entropy (BetaDCE), for feature selection. BetaDCE tries to estimate probability density of prediction using beta distribution, and it selects feature subsets by cross-entropy that maximizes its generalization ability. In Section 2, we discuss the basic theory. The results of feature selection by BetaDCE and the discussion are given in Section 3 and Section 4, respectively.

## 2. Materials and Methods

### 2.1. Beta Distribution-Based Cross-Entropy

The main objective of BetaDCE is feature selection, although it can also be used for prediction. Like KNN, BetaDCE tries to make full use of the information of adjacent samples. However, the characteristic of BetaDCE differing from KNN is that the probability density in the neighborhood is precisely modeled by beta distribution.

To evaluate feature subsets by BetaDCE, each training sample corresponds to an optimized number of nearest neighbors (*K*) which may differ from that of other samples. The parameter *K* is optimized by beta distribution *K*, and the expected value of probability is then used to compute the cross-entropy as the evaluation function for feature selection (Figure 1).

#### 2.1.1. Evaluation Function of BetaDCE

Suppose in a binary classification problem there are *K* nearest neighbors for the investigated sample, where *K_A_* and *K_B_* nearest neighbors belong to class *A* and *B* respectively. Although other definitions such as Mahalanobis distance are available, Euclidean distance is used here to define the nearest neighbors for simplicity. Let *P_A_* be the probability of the sample belonging to class *A*. It can be concluded that *P_A_* conforms to beta distribution with shape parameters of *K_A_* + 1 and *K_B_* + 1 in case no prior knowledge is provided. The expected value for *P_A_* can be analytically expressed as
(1)$$E[P_{A}]=\frac{K_{a}+1}{K_{a}+K_{b}+2}$$


Similarly, the equation of the expected value for *P_B_* can also be obtained. Therefore, the expected value (*E*) for the investigated sample can be written as
(2)$$E=\{\begin{array}{ll} E[P_{A}], & E[P_{A}]\geq E[P_{B}]\\ E[P_{B}], & E[P_{A}]<E[P_{B}] \end{array}$$


The overall cross-entropy for all the training samples can be calculated using the following formula:
(3)$$H=-\frac{1}{n_{A}}\sum_{i=1}^{n_{A}}\ln E_{i}-\frac{1}{n_{B}}\sum_{i=1}^{n_{B}}\ln E_{i}$$
where *E_i_* is the expected value for sample *i*, while *n_A_* and *n_B_* are the total numbers of samples for classes *A* and *B*, respectively. Equation (3) is used as the evaluation function of BetaDCE to optimize feature subsets.

#### 2.1.2. Iteration and Optimization

In the procedure of BetaDCE, a feature subset is initialized and then updated iteratively until a stopping criterion is reached. We define an epoch as a circle of iteration. Epoch *i* means that all subsets consisting of *i* features in the training dataset have had an opportunity to update the optimized feature subset.

In epoch *i*, all subsets consisting of *i* features are evaluated and then sorted by the values of the evaluation function defined by Equation (3). If the minimum value of the evaluation function in the current epoch is not smaller than 0.95 of that in the previous one, we update the optimized subset with one in the current epoch and exit the iteration. Otherwise, we update the optimized subset with the first *m* unique features from the list of sorted feature subsets and go to the next epoch. The number of unique features *m* is determined as follows:
(4)$$\begin{array}{cc} Max & m\\ s.t. & \frac{m!}{(i+1)!\left(m-i-1\right)!}\leq N_{e} \end{array}$$
where *i* is the index of epoch, and *N_e_* is the maximum number of feature subsets to be evaluated in every epoch, which is a predefined constant to control the amount of computation.

### 2.2. Investigated Datasets

Three different kinds of datasets were investigated to test the performance of BetaDCE. Firstly, an exclusive or-like (XOR-like) simulated dataset was used to test the ability for detecting redundant features by BetaDCE. Then, BetaDCE was used for feature selection in the training set of a leukemia dataset and prediction of class labels in the independent test set. Finally, in a metabonomic dataset, we applied BetaDCE to determine the number of features, which was extracted from raw features by partial least square (PLS) and used to predict whether chemotherapy would yield enough of a positive response on patients with gastric cancer.

To intuitively demonstrate its performance in feature selection, the model was tested on an XOR-like dataset. A raw training set containing 280 samples was generated for binary classification, where each class consisted of 140 samples. Samples of the two classes were crossed in four clusters in an XOR-like style, with normal distribution for samples in each cluster, to keep it non-linear. The raw training set was described by two features, and an additional redundant feature was then added, which was repeated 100,000 times to generate multiple training sets, with each containing one redundant feature, and the raw training set.

The leukemia dataset was taken from a collection of leukemia patients reported by Golub et al. [34]. The goal of this study was to develop a systematic approach to distinguish acute lymphoblastic leukemia (ALL) from acute myeloid leukemia (AML) based on the simultaneous measure of thousands of gene expressions using DNA microarrays. The training set consisted of 27 ALL and 11 AML samples. An independent collection of 34 leukemia samples (20 ALL and 14 AML) was used for testing. Each sample was described by 7129 probes from 6817 human genes.

The metabonomic dataset used in this study was about the chemosensitivity prediction of cisplatin plus 5-fluorouracil in a human xenograft model of gastric cancer [35]. In total, 9265 raw features were generated from the HPLC/MS profile and then extracted by PLS for prediction. There were 60 patients in the dataset, where 17 were sensitive to chemotherapy and the remaining were considered as the insensitive group.

### 2.3. Algorithm Implementation

We developed a Python implementation of BetaDCE, which takes advantage of multiple central processing unit (CPU) cores to significantly accelerate the training process, and the open source code is now available [36].

Our server had 80 Xeon E7-8870 CPU cores (2.10 GHz) and 120 GB of memory. The training time of BetaDCE for the leukemia dataset was 149.6 min on a single CPU, and it was shortened to 4.1 min when 80 CPUs were used. As a comparison, the training time of RF + sequential floating forward selection (SFFS) for the leukemia dataset was 380.3 min.

## 3. Results

### 3.1. Generalization Ability of BetaDCE

The generalization ability of a supervised learning method is related to both bias and variance. Higher bias leads to underfit, while higher variance leads to overfit. As for BetaDCE, the expectation value *E* combines the information of both bias and variance and can be a criterion for the evaluation of generalization ability.

Suppose there are two potential numbers of nearest neighbors, K1 = 4 and K2 = 8, to be determined. Of the four neighbors for K1, three are from the positive class, and one is from the negative class. For K2, six are from the positive class, and two are from negative class (Figure 2A). K1 and K2 obviously have the same bias toward the positive class (both are 3:1); however, the probability density of K2 is sharper, which results in lower variance than that of K1 (Figure 2B). *E* is the expectation value analyzed from the probability density to combine bias and variance. As a result, *E* for K2 is bigger than that for K1; therefore, we should choose K2 as the better number of nearest neighbors for better generalization ability.

By analysis of probability density, BetaDCE thereby combines information of both bias and variance in the model to ensure its generalization ability.

### 3.2. Analysis of XOR-Like Dataset

The XOR-like dataset was used to test the performance for feature selection. Here, BetaDCE was compared to KNN and SVM in terms of detecting the redundant feature in the dataset.

To compare the performance, the algorithms were firstly evaluated on the raw training set. Then, they were compared on the datasets with redundant features by the percentage of redundant features detected. For KNN, the accuracy of leave-one-out cross-validation was used to evaluate the performance of features, and the parameter *K* was optimized automatically in cross-validation. For SVM, the radial basis function kernel was used as the kernel function. The penalty parameter *C* and kernel coefficient were optimized simultaneously in cross-validation. The accuracy of five-fold cross-validation was used to evaluate the performance of features. For BetaDCE, the objective function was directly used for evaluation without the procedure of cross-validation, which can be safely done without data leakage since our task here is to test whether the model can identify the redundant features correctly or not, rather than to make a prediction in a test set.

Figure 3 shows the distribution of *E* on the raw two-dimensional data without additional redundant features during the training of BetaDCE. *E* clearly served as a satisfactory boundary for classification. It should be noted that the number of nearest neighbors of each sample can be determined automatically in an easy way during training; therefore, no other parameters are needed for training BetaDCE. The number of nearest neighbors varied considerably from sample to sample, with a minimal value of one and a maximal value of 68, but most of them were smaller than 10 (Figure 4).

A permutation test was adopted to test the performance in detecting redundant features in the dataset. The redundant feature was randomly permuted 100,000 times to calculate the objective function (BetaDCE) or accuracy of cross-validation (KNN and SVM). In the test, all three methods were able to detect the raw two-dimensional features. In total, 722 and 28,809 redundant features were not detected by KNN (Figure 5B) and SVM (Figure 5C), respectively, while all redundant features were detected by BetaDCE (Figure 5A). The overall false discovery rates were 0.7%, 28.8%, and 0% for KNN, SVM, and BetaDCE, respectively (Table 1). To test the superiority of BetaDCE to KNN and SVM, we calculated the false discovery rate every 1000 permutations for all the methods and set the null hypothesis that the false discovery rate of BetaDCE was not better than that of KNN (SVM). As a result, both null hypotheses for KNN and SVM were rejected. The false discovery rate of BetaDCE was significantly smaller than that of KNN (*p* = 3.8 × 10^−21^) or SVM (*p* = 5.7 × 10^−124^), which indicated that the performance of BetaDCE for feature selection was much better than that of KNN and SVM with cross-validation.

### 3.3. Analysis of Leukemia Dataset

#### 3.3.1. Predictive Result of BetaDCE

As described previously, 38 samples of leukemia were analyzed to test whether it was possible to classify them as ALL or AML based on 7129 genes using DNA microarrays. An independent test set of 34 leukemia samples was then used to evaluate the performance of the classification model trained. The performance of BetaDCE was compared with Golub’s method, as well as least absolute shrinkage and selection operator (LASSO), and random forest (RF) with sequential floating forward selection (SFFS).

In Golub’s work [34], a total of 50 genes most closely correlated with AML–ALL distinction in the training set were chosen as the set of informative genes. The class predictor was based on “weighted votes” of the informative genes. Before training of BetaDCE, all the genes were standardized to unit variance by subtracting the mean and then dividing by the standard deviation. The evaluation function decreased during the first few epochs, and the stopping criterion was reached at the fourth epoch (Figure 6). Figure 7A,B show the bias and variance trade-off for BetaDCE on the leukemia dataset. As can be seen, the variance was dominant when the number of nearest neighbors was small, and the bias became the main contribution to predictive results for big numbers of nearest neighbors. The cross-entropy by BetaDCE is a trade-off between bias and variance. The area under the curve (AUC) of BetaDCE on the test set was 0.93, which was larger than that of Golub’s method (0.83), RF + SFFS (0.81), and LASSO (0.81). The comparison between BetaDCE and other methods is shown in Table 2, which indicated that BetaDCE could provide higher predictive performance with a much smaller number of selected features.

#### 3.3.2. Interpretation of the Features Selected by BetaDCE

One advantage of feature selection is that it eases interpretation of the system revealed by data. For the leukemia dataset, BetaDCE finally selected four genes which were crucial for the distinction of AML and ALL. The accession numbers of selected genes were M54995_at, M84526_at, X16662_at, and M22612_f_at. Detailed information about the selected genes is shown in Table 3.

Furthermore, we measured the predictive power of each input feature by measuring the area under the curve (AUC) of the receive operating characteristic (ROC) curve. Figure 8 shows the AUCs of all the features in the leukemia dataset. From the figure, we can see that, among the four genes selected by BetaDCE, AUCs of M54995_at (0.95) and M84526_at (0.92) were high, whereas AUCs of X16662_at (0.50) and M22612_f_at (0.51) were at the average level.

### 3.4. Analysis of the Metabonomic Dataset

The metabonomic dataset was randomly split into a training set (2/3) and a test set (1/3), as described by Wang et al. [35]. The splits of the dataset were repeated 1000 times, and predictions on test sets were evaluated by ROC curves. Raw features were generated by the HPLC/MS profile and extracted by PLS. Then, BetaDECE or cross-validation was used to determine the number of features for predictive model. As indicated in Figure 9, the average true positive rate by BetaDCE was higher than that by cross-validation for most points of false positive rate. A paired comparison by AUC indicated that the performance of feature selection by BetaDCE was significantly better than that by cross-validation (*p* = 2.4 × 10^−112^).

## 4. Discussion

Cross-entropy is widely used as a loss function in machine learning, such as logistic regression and deep neural networks. It minimizes the difference between the estimated probability distribution and the true distribution from the perspective of information theory. The estimated probability distribution needed for cross-entropy is generally predicted from the training samples. However, the prediction of probability distribution from the limited number of samples could be much more difficult than the original problem. The variance may be high, especially when the number of samples is small compared to the number of features.

In view of such difficulties, BetaDCE uses beta distribution as the estimated probability distribution to calculate cross-entropy. Meanwhile, the expected value of beta distribution is analytically available; thus, estimation of cross-entropy is easy. Therefore, BetaDCE is efficient because no resampling methods such as cross-validation are needed, and the parameter *K* is optimized automatically during training without the requirement of additional parameters for optimization.

The limitation of BetaDCE is that, when the number of features in the investigated dataset is huge compared to number of samples, the solution might be biased from the optimal one. Under such circumstances, we should enlarge the constant Ne to increase the probability of getting to the optimal solution at the expense of greater computational effort. This limitation is essentially the problem of wrapper methods.

## 5. Conclusions

In summary, we proposed BetaDCE as a novel approach for feature selection. The characteristic of BetaDCE is that the cross-entropy of a predictive model is computed directly by the expected value of beta distribution, so that the generalization ability can be estimated more precisely than conventional methods such as cross-validation. We analyzed the generalization ability of BetaDCE by trade-off between bias and variance. The performance of BetaDCE was validated on several datasets. In the XOR-like dataset, the false discovery rate of BetaDCE was significantly smaller than that of other methods. For the leukemia dataset, the AUC of BetaDCE on the test set was 0.93 with only four selected features, indicating that the performance of prediction by BetaDCE can be ensured with a smaller number of features than the original method, whose AUC was 0.83 with 50 features. In the metabonomic dataset, the overall AUC of the prediction with features selected by BetaDCE was significantly larger than that by the original reported method.

Although the robustness of BetaDCE was validated in this study, its limitation is that the optimal solution cannot be promised if there are a huge number of features. In future work, one practical way of ensuring the optimal solution is to implement the algorithm on a graphics processing unit (GPU) for efficient parallelization. Another way is to organize the algorithm of BetaDCE as a constrained mathematical program, so that it can be solved as an embedded method.

## Figures and Tables

**Figure 1 entropy-21-00769-f001:**
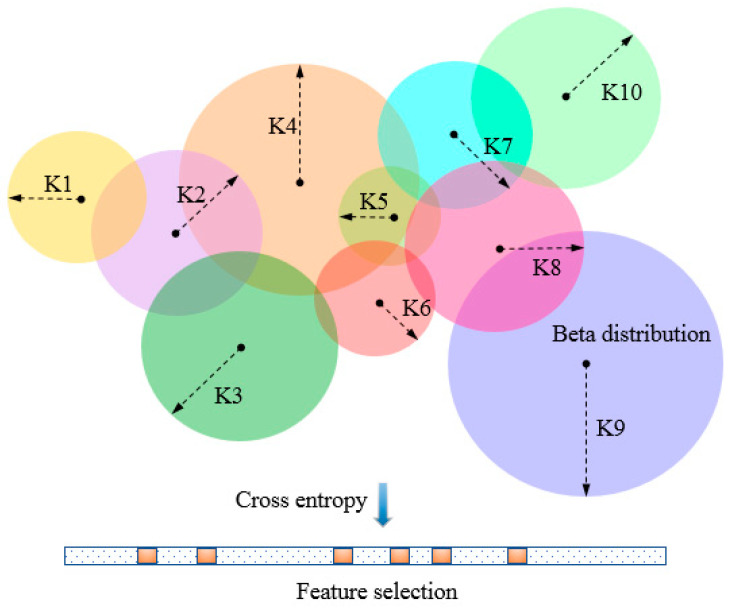
Description of beta distribution-based cross-entropy (BetaDCE). Each training sample corresponds to an optimized *K*, which is determined by beta distribution. The expected value of probability is then used to compute the cross-entropy as the evaluation function for feature selection.

**Figure 2 entropy-21-00769-f002:**
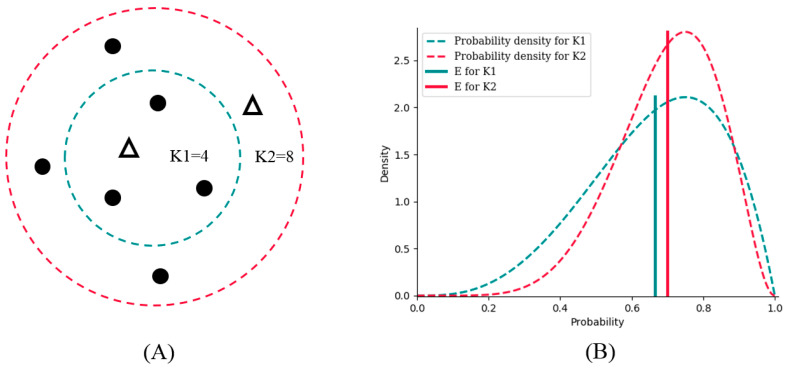
The generalization ability of BetaDCE. (**A**) An example comparing two numbers of nearest neighbors. K1 and K2 are two potential numbers of nearest neighbors, with the same bias, to be determined. (**B**) The probability density and *E* (expected value) for K1 and K2. Although the biases of K1 and K2 are the same, the probability density and *E* indicate that we should choose K2 for better generalization ability.

**Figure 3 entropy-21-00769-f003:**
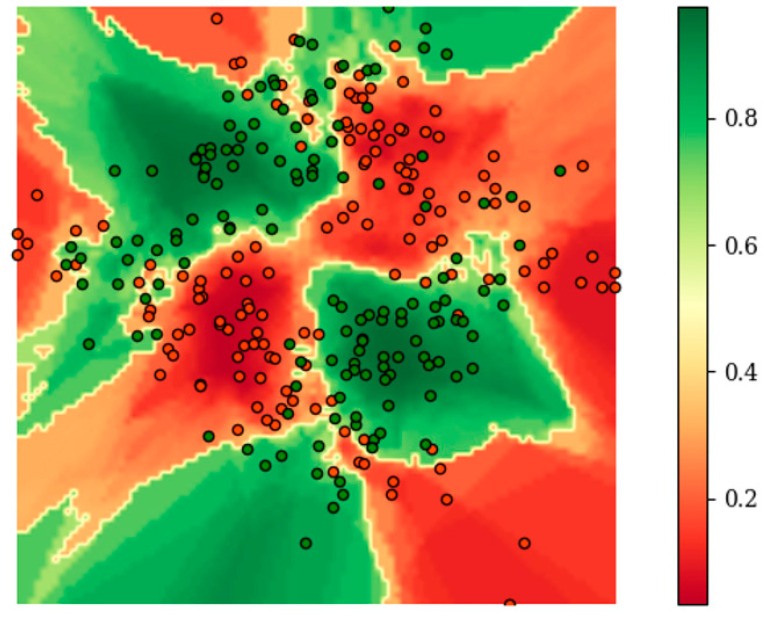
Distribution of *E* on the raw two-dimensional data of the exclusive or-like (XOR-like) dataset during training of BetaDCE.

**Figure 4 entropy-21-00769-f004:**
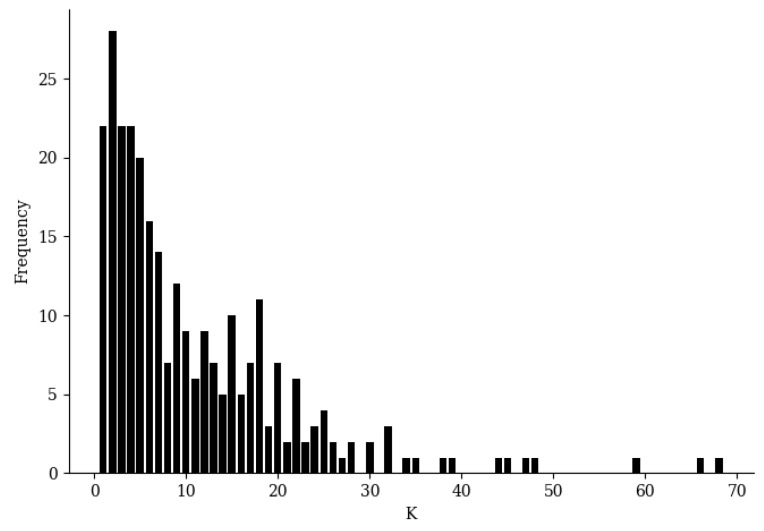
Statistics of number of nearest neighbors (K) for BetaDCE training on the raw XOR-like dataset without redundant features.

**Figure 5 entropy-21-00769-f005:**
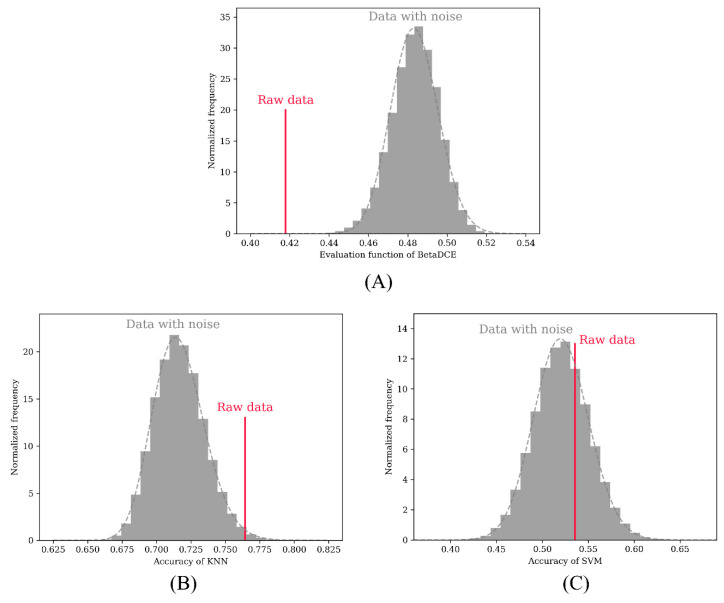
Permutation test for detecting redundant features on the XOR-like dataset. (**A**) Permuted result of BetaDCE. The statistics were based on the evaluation function. (**B**) Permuted result of k-nearest neighbor (KNN). The statistics were based on the accuracy of leave-one-out cross-validation. (**C**) Permuted result of support vector machine (SVM). The statistics was based on the accuracy of five-fold cross-validation.

**Figure 6 entropy-21-00769-f006:**
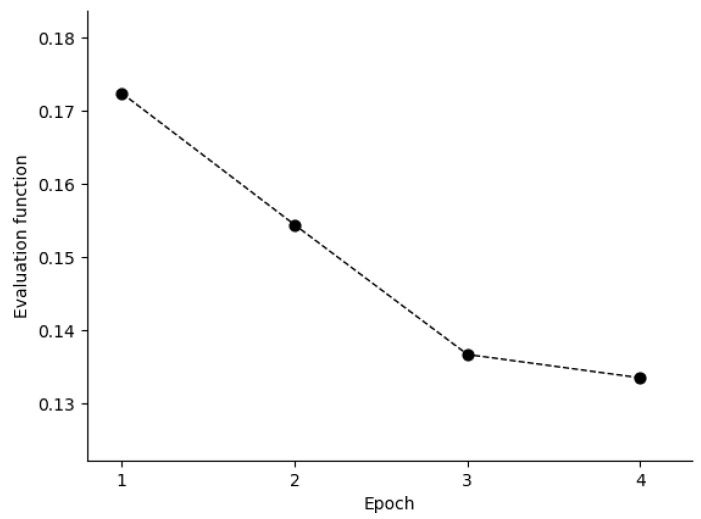
Relationship between evaluation function of BetaDCE and training epoch. Epoch *i* means that all subsets consisting of *i* features in the training dataset have had an opportunity to update the optimized subset.

**Figure 7 entropy-21-00769-f007:**
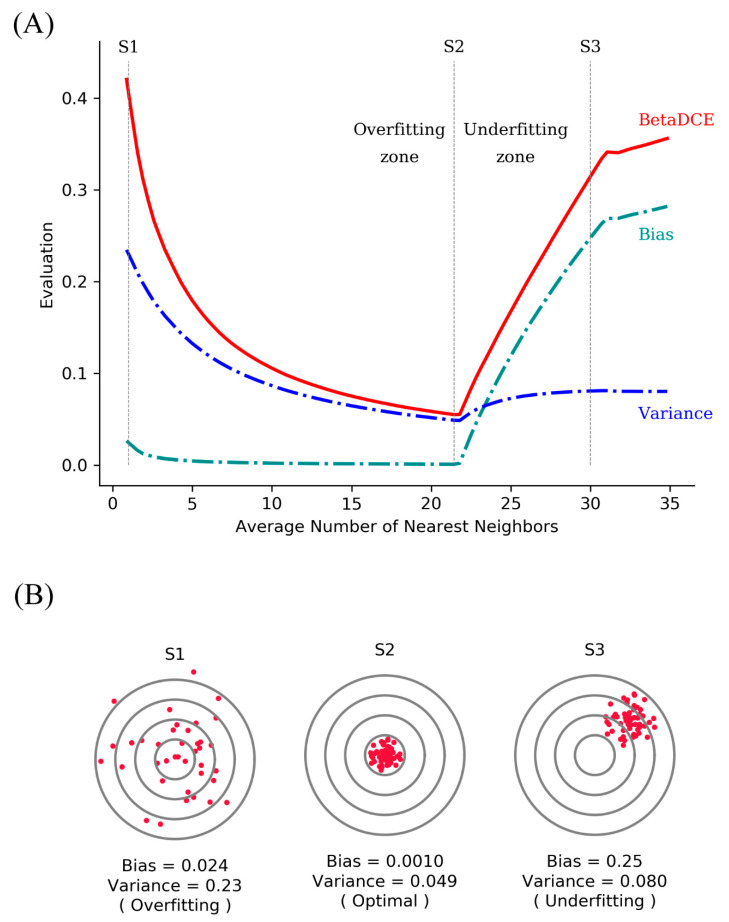
Bias and variance trade-off for BetaDCE on leukemia dataset. (**A**) Relationship among the evaluations of BetaDCE, bias, and variance. Here, this was the bias and variance trade-off when the number of nearest neighbors was optimized for the optimal feature subset. Variance was shown in the form of standard deviation so that it was comparable with bias and the value of BetaDCE. Red solid line: evaluation by BetaDCE. Blue dashed dotted line: bias. Green dashed dotted line: variance. The average numbers of nearest neighbors for S1, S2, and S3 were one, 21.4, and 30 respectively. S2 was the optimal parameter dividing the whole space into overfitting zone (left) and overfitting zone (right). (**B**) The bulls-eye diagrams for customized points S1, S2, and S3. Data were simulated according to the size of bias and variance.

**Figure 8 entropy-21-00769-f008:**
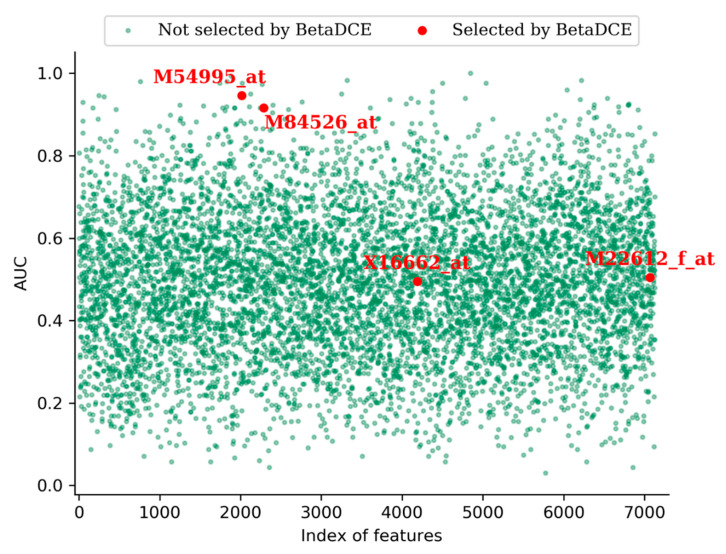
Predictive power of the features in the leukemia dataset. The predictive power of a feature is measured by the area under the curve (AUC). Red points are the genes selected by BetaDCE, and the corresponding gene accession numbers are shown. The order of features listed in the figure is the same as that in Golub’s work.

**Figure 9 entropy-21-00769-f009:**
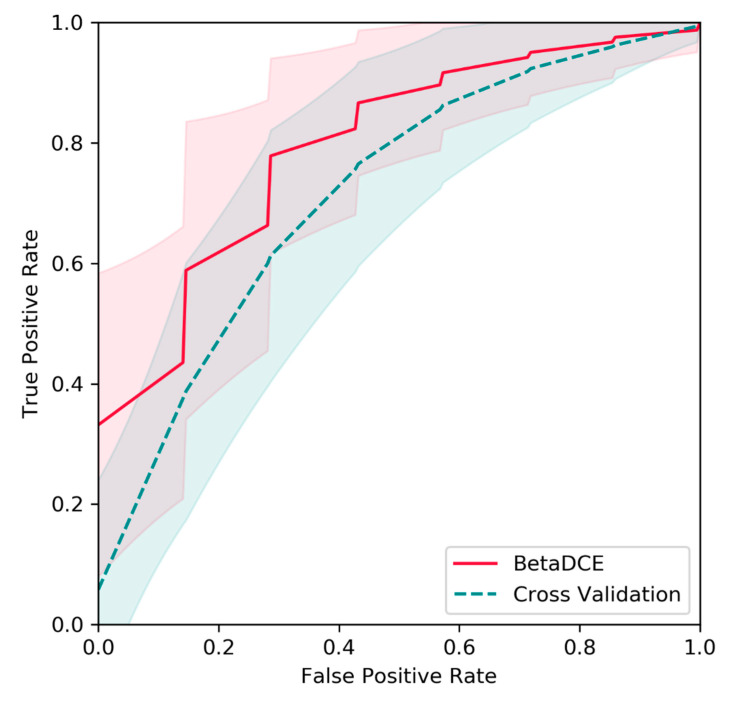
Comparison of BetaDCE and cross-validation for feature selection on the metabonomic dataset. The predictive results based on 1000 random splits were compared in terms of receive operating characteristic (ROC) curves. The red and blue filled areas are the standard deviations for BetaDCE and cross-validation, respectively.

**Table 1 entropy-21-00769-t001:** Comparison between beta distribution-based cross-entropy (BetaDCE) and other methods for detecting redundant features on an exclusive or-like (XOR-like) dataset. KNN—k-nearest neighbor; SVM—support vector machine.

Methods	Evaluation Function	False Discovery Rate (1-Specificity)
KNN	Cross-validation	0.7%
SVM	Cross-validation	28.8%
BetaDCE	Equation (3)	0%

**Table 2 entropy-21-00769-t002:** Comparison between BetaDCE and other methods on the leukemia dataset. RF—random forest; SFFS—sequential floating forward selection; LASSO—least absolute shrinkage and selection operator; AUC—area under the curve.

Methods	AUC on Test Set	Number of Selected Features
Reported by Golub [34]	0.83	50
RF + SFFS	0.81	3
LASSO	0.81	111
BetaDCE	0.93	4

**Table 3 entropy-21-00769-t003:** Description of genes selected by BetaDCE.

Gene Accession Number	Description of the Gene
M84526_at	DF: D component of complement (adipsin)
X16662_at	ANX8: Annexin VIII
M54995_at	PPBP: Connective tissue activation peptide III
M22612_f_at	PRSS1: Protease; serine; 1 (trypsin 1)

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
