# Peer review of "Beta Distribution-Based Cross-Entropy for Feature Selection"

_entropy, 2019, doi:10.3390/e21080769_

Round 1

Reviewer 1 Report

General comments:

While the authors have put significant effort to solve a significant problem, the presentation and structure of the work can be improved. My main concerns is that there is no evidence of significant literature review of related work, no clear statements of limitations of this work, no conclusion section, no future work section, stating how to address the limitations. Other minor issues are mentioned below.

The results of the experiments should be added in brief in the abstract

In lines 20-21, “it also more accurately predicted the class labels with smaller number of features.”.

Please state by what percent is more accurately predicted.  

 The work is based on past research/results [26],[27] conducted on 2006 and 1983 and [25] published on 2011. Is there a more recent attempt. If available, referring to more recent work will make the paper more convincing that it addresses a problem that has not yet been addressed.

The paper can benefit by the addition of a “related work” section.

Line 126, “at: https://github.com/mlalgorithm/betadce.”, consider replacing with a reference.

In figure 7, the labels of the red points are difficult to read. Consider making them bold.

Limitations of this work are not mentioned by the authors.

Consider adding a conclusions and limitations section.

Reviewer 2 Report

The paper is very interesting and it will be of interest for researchers in many different areas such as applied computer science, statistics, food science, etc

Reviewer 3 Report

What is the meaning of BDCEFS authors mentioned in the abstract? It is not the same the version placed at the manuscript against the one placed at the Review Report Form. 

It is vague to said that algorithm implementation takes advantage of multiple CPU cores to significantly accelerate the training process. Which is the minimum of computing power required to have good results with the technique reported in the manuscript? How this can be mensure against other wrapper methods?

Bulls-eye diagram with both bias and variance can help to see the results of the algorithm implementation over the data sets involved, so it is suggested to do that in orden to show the bias and variance trade-off complexity of the algorithm applied to the datasets involved.

It is encourage to present a benchmark of wrappers methods for feature selection to demostrate the outperformance describes in the manuscript.

Reviewer 4 Report

Given the increasing prevalence of high-dimensional datasets, feature selection remains an important topic. Presented in this article is an interesting method that uses a beta distribution based cross entropy to perform feature selection. Due to this approach, the method can also serve as a simple classification model, in addition to its feature selection capabilities. Nevertheless, I do have some comments about the article.

1.       For equation 1, it doesn’t appear that “nearest neighbors” is defined.

2.       It’s not that clear why the XOR-like dataset has 280 samples. What was the reason for this choice?

3.       For the XOR-like dataset, it’s unclear how many features were generated. From the figure, it appears like two, but this should be clarified.

4.       On the XOR-like dataset, KNN is used as a comparison method, but it’s unclear how this was done. KNN is not a typical feature selection method, and I would suggest using a more mainstream method if there is not a good reason for this.

5.       For the XOR-like dataset, the comparison appears to be only of the sensitivity of the two methods, but what about the specificity?

6.       For the XOR-like dataset, The manuscript states that for BetaDCE the objective functions was directly used for evaluation, without cross-validation. However, BetaDCE depends on an iterative optimization. If this is so, that seems like it is being tested on the same data it is trained on, which is not a fair evaluation. Please clarify this.

7.       The interpretation of the p-value for the permutation test is not correct (line 179). The authors did not test of the hypothesis of superiority.

8.       The leukemia dataset used is very small. There are much larger datasets available these days. This means that small overall differences in classification will have a larger effect on accuracy.

9.       Compounding the problems with the small sample size with the Golub data is that the authors are comparing to a method that was applied 20 years ago for a performance comparison. Clearly, much has happened in the intervening time. The paper would be stronger if an evaluation using more current methods were used, or if at least a mainstream method (e.g. LASSO) were applied to the datasets to get a comparison.

10.   What is meant by predictive results is not really defined in Table 2. If this is accuracy, what kind of accuracy? AUC would probably be a better choice, it is easy to interpret and less problematic that classification accuracy.

11.   On lines 221-222: the predictive power of individual features may well be high, but the fact that some of them are low does not seem to indicate that genes may function cooperatively. Otherwise, one would expect the combined accuracy to be higher.

12.   In the Discussion, the authors mention that cross entropy makes the optimization of weights easier, but that is not related to the method being discussed.

Round 2

Reviewer 1 Report

Most of the comments have been addressed, however the conclusion requires some extra work. 

Please increase the size of the conclusion by summarizing the key points of the paper as well as the results. Consider adding a future work section in the conclusion, where you could explain how the limitations can be addressed. 

In some cases the writing can be improved, e.g "To overcome the problem, development of an embedded method 291 based on BetaDCE is the direction of our future work." 

Moderate English changes in various places in the paper with improve the quality and make the paper easier to read. 

Reviewer 3 Report

Dear authors,

Thank you for your effort, here you have the observations to the last revision:

- Graph 7 is key and a great way to explain the bias and variance trade-off, maybe axes values can be changed by others that provide another perspective of how the values reported can be used to minimize two source of error which prevent supervised learning algorithms from generalizing well to new data.

File attached provide graph examples to attend bias and variance trade-off results exhibition. In this way, underfitting, overfitting and prediction can be exhibited accordingly.

- Last revision of grammar and typos is required.

- Conclusions section must be enhanced.

Regards.

Reviewer 4 Report

Although the paper has certainly improved, I still have a few comments:

1. I suggest you define what is meant by nearest, for the nearest neighbors. There are different ways this can be measured.

2. When I first reviewed this manuscript, I was confused by what was meant by "features of noise". That is clearer now, but the wording might be difficult to understand. If you mean "true negatives" then I suggest trying to find a better way to say that.

3. The conclusions are still based on a relatively small dataset. I would suggest adding an additional dataset.
